# Hypoxia Promotes Angiogenic Effect in Extracranial Arteriovenous Malformation Endothelial Cells

**DOI:** 10.3390/ijms23169109

**Published:** 2022-08-14

**Authors:** Joon Seok Lee, Hyun Geun Cho, Jeong Yeop Ryu, Eun Jung Oh, Hyun Mi Kim, Suin Kwak, Seok-Jong Lee, Jongmin Lee, Sang Yub Lee, Seung Huh, Ji Yoon Kim, Ho Yun Chung

**Affiliations:** 1Department of Plastic and Reconstructive Surgery, School of Medicine, Kyungpook National University, Daegu 41944, Korea; 2Cell & Matrix Research Institute, School of Medicine, Kyungpook National University, Daegu 41944, Korea; 3BK21 FOUR KNU Convergence Educational Program of Biomedical Science for Creative Future Talents, Department of Biomedical Science, School of Medicine, Kyungpook National University, Daegu 41944, Korea; 4Department of Dermatology, School of Medicine, Kyungpook National University, Daegu 41944, Korea; 5Department of Radiology, School of Medicine, Kyungpook National University, Daegu 41944, Korea; 6Department of Surgery, School of Medicine, Kyungpook National University, Daegu 41944, Korea; 7Department of Pediatrics, School of Medicine, Kyungpook National University, Daegu 41944, Korea

**Keywords:** arteriovenous malformation, endothelial cell, hypoxia, angiogenesis

## Abstract

Arteriovenous malformation (AVM) is characterized by high-flow blood vessels connecting arteries and veins without capillaries. This disease shows increased angiogenesis and a pathophysiological hypoxic environment in proximal tissues. Here, we analyzed the effects of hypoxia on angiogenesis in the endothelial cells (ECs) of AVM and normal tissues. ECs from human normal and AVM tissues were evaluated using immunocytochemistry with CD31. In vitro tube formation under hypoxia was tested in both ECs using Matrigel. The relative expression of angiogenesis-related genes was measured using real-time PCR. Under normoxia, CD31 was significantly higher in AVM ECs (79.23 ± 0.65%) than in normal ECs (74.15 ± 0.70%). Similar results were observed under hypoxia in AVM ECs (63.85 ± 1.84%) and normal ECs (60.52 ± 0.51%). In the tube formation test under normoxic and hypoxic conditions, the junction count and total vessel length were significantly greater in AVM ECs than normal ECs. Under both normoxia and hypoxia, the angiogenesis-related gene *FSTL1* showed a significantly higher expression in AVM ECs than in normal ECs. Under hypoxia, *CSPG4* expression was significantly lower in AVM ECs than in normal ECs. Accordingly, the angiogenic effect was increased in AVM ECs compared with that in normal ECs. These results provide a basic knowledge for an AVM treatment strategy.

## 1. Introduction

Arteriovenous malformation (AVM) is a rare disease characterized by arteries that are directly connected to draining veins with a high flow rate, without an intervening capillary network. Arteriovenous shunting without an intervening capillary network leads to the formation of an abnormal vascular network around the nidus as new blood vessels are formed. Extracranial AVM can be caused by various factors such as hormonal changes, infections, and trauma, and can occur in any body part [1]. Although AVM can be minor and asymptomatic, it may develop into a fatal disease. Extracranial AVM is typically diagnosed using vascular Doppler ultrasonography or computed tomography angiography.

AVM is pathophysiologically related to angiogenesis, which may be due to hypoxia or ischemia conditions. Although the exact cause is unknown, AVM has the pathologic shunt congenitally. As AVM progresses, high-pressure arterial blood is directly shunted into low-pressure veins, dilating the vessels and recruiting new blood vessels. Such continued stimulation increases the risk of hemorrhage and bleeding, deteriorated flow can cause blood steal phenomena, and the circulation of high-flow blood creates a relatively hypoxic environment in the proximal tissues at the AVM site. Patients with AVM often have a high hemorrhage risk and may require several surgeries along with embolization and sclerotherapy [2,3,4].

Anti-angiogenic drugs such as vascular endothelial growth factor (VEGF) blockers are also used to prevent the vicious cycle of regeneration by stimulating angiogenesis, as various dynamic entities occur during AVM treatment. Rapamycin, which is the mammalian target of the rapamycin (mTOR) inhibitor and also known as sirolimus, is used for AVM treatment, and trametinib, the MEK1 inhibitor, receives attention as AVM treatment nowadays. Aside from this, non-steroid anti-inflammatory drugs, analgesics, and low molecular weight heparin are used as the treatment of AVM patients. [5,6,7,8,9,10,11,12,13]. Moreover, various studies have reported that angiogenesis induced under hypoxia was effectively compared and evaluated in normal and pathophysiological situations using the CD31 expression [14,15].

To understand and treat AVM, its mechanism and physiopathology should be evaluated, and the related genes should be analyzed to develop therapeutic strategies. In this study, we investigated the effects of hypoxia on endothelial cells (ECs) from AVM tissues and assessed the expression of related genes.

## 2. Results

### 2.1. Immunofluorescence

Immunocytochemical fluorescence tests were performed on ECs that were isolated and cultured from human normal and AVM tissues. A green color was observed on the cell membrane, indicating CD31-positive staining. These results confirmed that the cultured and isolated cells were ECs (Figure 1).

The comparison of CD31 expression in normal ECs and AVM ECs was assessed using immunofluorescence microscopic image analysis after incubation under each condition (normoxia and hypoxia) for 14 h. Under normoxia, CD31 immunoreactivity was found in 79.23 (±0.65%) of the area in AVM ECs, which was greater than that in normal ECs (74.15 ± 0.70%). Similarly, under hypoxia, CD31 immunoreactivity was found in 63.85 (±1.84%) of the area in AVM ECs, which was significantly higher than that in normal ECs (60.52 ± 0.51% in normal ECs), suggesting that angiogenesis is more active in AVM ECs (Figure 2).

### 2.2. Tube Formation Assay

The in vitro tube formation effect of hypoxia was examined using Matrigel for both normal and AVM ECs. After 8 h of incubation, the number of junctions and total vessel length increased significantly in AVM ECs compared with that of normal ECs under normoxia (*p* < 0.05). Similarly, under hypoxia, the number of junctions and the total vessel length increased significantly in AVM ECs compared to those in normal ECs (*p* < 0.05). Additionally, both normal and AVM ECs showed a significantly increased number of junctions and total vessel lengths under hypoxia than under normoxia (Figure 3 and Figure 4).

### 2.3. Real-Time PCR

The relative expression levels of the three target genes, *FSTL1*, *CSPG4*, and *MARCKS*, which were selected from differentially expressed genes between normal ECs and AVM ECs through NGS (next generation sequencing) (data was not shown), were measured using real-time PCR. Under both normoxia and hypoxia, *FSTL1* expression was significantly higher in AVMs than in normal tissues. Under hypoxia, *FSTL1* expression was significantly increased in the ECs of both AVM and normal tissues. *CSPG4* expression was significantly lower in AVM ECs than in normal ECs under normoxia, because the capillary network was absent in AVM. In particular, under hypoxia, *CSPG4* expression decreased significantly in AVM ECs compared to that in normal ECs. *MARCKS* expression was lower in AVM ECs than that in normal ECs under both normoxia and hypoxia. Although there was no marked difference in the expression levels of *MARCKS* between normoxia and hypoxia, *MARCKS* expression was significantly decreased in normal ECs and AVM ECs (Figure 5).

## 3. Discussion

Arteriovenous malformation is a rare vascular malformation that can cause vascular enlargement over time and is associated with high morbidity and a risk of recurrence after treatment. The physiopathologic characteristics of AVM include a fistula shunted from the arteries to the veins without an intervening capillary network bed, vascular malformation with channel formation in the fistula or feeding arteriole resulting in a nidus that connects to the draining vein, and bleeding or hemorrhage caused by high-pressure flow. Such fast flow shunting generates relative hypoxia in the normal proximal tissues, even under normoxia. We investigated the effects of hypoxia on ECs and the expression of related genes. Several previous studies have reported neovascularization in AVM. AVM has been described as lesions enlarged through vasculogenesis, involving the formation of new vasculature or angiogenesis to form new blood vessels from the pre-existing vasculature [16,17,18,19,20,21,22,23,24]. First, the effect of EC proliferation on angiogenesis was evaluated [14,15] and was found to be greater in AVM than in normal tissues. EC proliferation increased in AVM compared to normal tissues under hypoxic conditions, although proliferation generally decreased because of hypoxic damage (Figure 1 and Figure 2). An increased tube formation was associated with an increased number of junctions and total vessel lengths, suggesting that angiogenesis was enhanced. Both the number of tube junctions and the total vessel lengths were significantly higher in AVM ECs than in normal tissues. In addition, under hypoxia, greater tube formation was observed in AVM ECs than in normal ECs, suggesting greater angiogenic patency in AVM than in normal tissues (Figure 3 and Figure 4).

Among the various genes associated with angiogenesis, we profiled *FSTL1*, *MARCKS*, and *CSPG4*. The expression of all three genes was higher under hypoxia than under normoxia. Specifically, *FSTL1* expression was significantly increased in AVMs under pathophysiological hypoxic conditions. This suggests that *FSTL1* is closely associated with AVMs. In a previous study, *FSTL1* stimulated angiogenesis in smooth muscles under hypoxic conditions and increased cardiac hypertrophy [25]. Increased FSTL1 levels have also been reported to be protective against ischemic damage in skeletal muscles. FSTL1 suppresses smooth muscle cell proliferation through an AMPK-dependent mechanism and attenuates neointimal formation in response to arterial injury [26]. In addition, *FSTL1* affects the proliferation of breast cancer cells and vascular ECs involved in angiogenesis [27]. Consistent with these findings, *FSTL1* was significantly increased in AVM ECs under pathological hypoxia compared to normal ECs to protect proximal tissues against hypoxic damage, and hypoxia increased *FSTL1* expression in AVMs compared to that in normal tissues (Figure 5).

*CSPG4* expression is regulated by inflammation and hypoxia-induced signal transduction. In particular, under normoxia, *CSPG4* expression was detected only in the pericytes of the arterioles and capillaries and not in the venules [28,29,30]. AVM is characterized by the physiological absence of capillaries, which leads to fast shunting from the arteries to the veins, resulting in pressure and hypoxic conditions in the proximal tissues. Capillaries, which occupy the largest area in the normal blood system, were unaffected by AVM, and *CSPG4* expression was significantly lower in AVM ECs than in normal ECs. When hypoxic conditions were induced, *CSPG4* expression was increased in normal ECs and capillary networks formed. In contrast, in AVM ECs, *CSPG4* expression was only increased in the arterioles. Although *CSPG4* expression was increased under hypoxia compared to normoxia, the expression level was lower in AVM ECs than in normal ECs (Figure 5). Therefore, among the genes increased under hypoxic conditions, *CSPG4* may be an important factor for distinguishing AVM from other diseases. Our findings suggest that the ECs in AVM lack the capacity to form capillary networks.

*MARCKS* was recently shown to be related to angiotensin II in kidney cancer and to play a key role in neo-angiogenesis. In *MARCKS*-knockdown xenograft tumors, microvessels showed the lowest density of the endothelial marker CD31 [31]. Similarly, we found that the induction of hypoxia led to a marked increase in *MARCKS* expression in normal ECs. However, under normoxia, *MARCKS* expression was lower in AVM ECs than in normal ECs. Under hypoxic conditions, *MARCKS* expression was slightly increased in AVM ECs, but did not significantly differ from that in AVM ECs under normoxia. As described by Zhao et al., who analyzed the role of *MARCKS*, hypoxia activates HIF-1α, leading to various mechanisms that induce angiogenesis, including the release of cytokines, such as VEGF. However, *MARCKS* is not related to VEGF and increases its phosphorylation [32]. These findings suggest that *MARCKS* is not strongly related to AVM despite its association with angiogenesis (Figure 5).

The distinct pathophysiology of AVM, involving direct shunting with high pressure from the arteries to the veins, differs from the features of normal vessels. Further induction of hypoxic conditions caused changes in gene expression. As AVM does not affect capillary networks, we predicted that *CSPG4*, which is mainly expressed in the capillaries, would decrease. As expected, *FSTL1* expression was increased under hypoxic conditions. However, *MARCKS* expression was increased to protect against hypoxic damage after hypoxic conditions were simulated. Although further studies are needed to investigate the pathways directly connected to *MARCKS* in AVM, cytokines such as VEGF that are produced under hypoxia may be related to the lower increase in *MARCKS* expression under hypoxia than under normoxia. This finding also suggests that *MARCKS* is not associated with AVM.

## 4. Materials and Methods

### 4.1. Isolation and Culture of ECs

This study was approved by the Institutional Review Board of Kyungpook National University Hospital (approval number: KNUH 2020-03-078-001) and was performed in accordance with the Helsinki Declaration. Ten patients (five with and five without AVM) were enrolled in this study. After obtaining patient consent, AVM tissues were collected during surgical resection of AVM lesions and normal arterial vasculature tissues were collected during other surgeries from discarded normal subcutaneous tissues, including normal arterial vasculature. Surgical samples were rinsed with phosphate-buffered saline. Tissues were cut into small pieces using a surgical scalpel and scissors. To separate the epidermis and dermis, tissues were incubated in 0.01% dispase II (Dispase 17105-041, Thermo Fisher Scientific, Waltham, MA, USA) for 24 h at 4 °C. After incubation, the dermal layer was collected and placed in 10 mL of Hank’s balanced salt solution (14170-112, Thermo Fisher Scientific, Waltham, MA, USA). The supernatant was removed, 5 mL of collagenase type I was added, and the samples were incubated at 37 °C and 170 bpm in a shaking incubator for 1 h. Endothelial cell growth basal medium-2 medium (10 mL; EBM-2 media CC-4176, Lonza, Walkersville, MD, USA) was added, and the sample was passed through a 70 μm nylon filter. After centrifugation (2000 rpm, 5 min), ECs were grown in EMB-2 medium and incubated at 37 °C with an oxygen level that was the same as that in air and 5% CO_2_. ECs were grown, subcultured, and then isolated using a CD31 Microbead kit (Mini&MidiMACS starting kit 130-042-501, Miltenyi Biotec, Gladbach Bergisch, Germany) [33].

### 4.2. Hypoxic Conditions

Hypoxic conditions used in the experiments consisted of 1% O_2_ and 5% CO_2_ at 37 °C. To establish hypoxic partial pressure, a hypoxia chamber (Modular Incubator Chamber, Billups-Rothenberg, Inc., San Diego, CA, USA) was used. Cultured ECs were exposed to hypoxic conditions for 14 h.

### 4.3. Immunofluorescence Imaging

For staining, cells were cultured in an eight-well chamber. When the cells reached approximately 70–80% confluence, they were fixed in 4% formaldehyde for 24 h, immersed in 0.3% Triton-X100 in Tris-buffered saline for 10 min, and rinsed three times with deionized water for 1 min. Ten drops of blocking buffer immunotech-normal serum were added to each well and allowed to react for 5 min. After removing the serum, the primary antibody, CD31 (ab28364, Abcam, Cambridge, UK, 1:100, 150 μL), was added to the wells and incubated at 4 °C for 24 h. After washing three times, the secondary antibody, goat anti-rabbit IgG H&L (Alexa Fluor^®^ 488 A-11008; Invitrogen, Carlsbad, CA, USA, 1:100), was added and incubated for 1 h. After washing with Tween-20 three times for 10 min each, Vectashield (H-1200 with DAPI) was added to the slide, and the slides were covered. Cells were evaluated using a fluorescence microscope (Carl Zeiss, Oberkochen, Germany). CD31 expression was quantified by measuring the percentage area of immunoreactivity using the ImageJ software (NIH, Bethesda, MD, USA).

### 4.4. Tube Formation Assay

To analyze angiogenesis, 50 μL of each ice-state extracellular matrix solution (Angiogenesis Assay Kit, in vitro, ab204726) was added to a 96-well plate and incubated at 37 °C for 1 h to form a gel. ECs were added at a density of 1 × 10^4^/mL and cultured for 14 h. Cells were stained with calcein fluorescent dye at 37 °C for 30 min and evaluated under a fluorescence microscope (AXIO, Carl Zeiss, Oberkochen, Germany). Angiogenesis was detected using photographs of blood vessels and anchorage junctions using AngioTool64 software (National Cancer Institute, Radiation Oncology Branch, Angiogenesis Core Facility, Bethesda, MD, USA) [34].

### 4.5. Real-Time PCR

Real-time PCR was performed to analyze gene expression in the cells after exposure to hypoxic conditions. First, RNA was extracted from each group using TRIzol and reverse-transcribed into cDNA using a reverse transcription premix (EBT-1515, Elpis Biotech, Daejeon, Korea). Reactions were performed using SYBR Green PCR Master Mix (1708882. Bio-Rad, Hercules, CA, USA) in a real-time PCR detection system (CFX96 Touch; Bio-Rad, Hercules, CA, USA). GAPDH was quantified in parallel, with the target gene as an internal control. Normalization and fold-changes were calculated using the Ct method (Table 1).

### 4.6. Statistical Analysis

Data are presented as the mean ± standard deviation of all experimental replicates. One-way ANOVA was used for multiple comparisons in experiments with one independent variable in Excel (Microsoft, Redmond, WA, USA). Statistical significance was set at *p* < 0.05.

## 5. Conclusions

Angiogenesis is increased in AVM, which is difficult to treat and is characterized by direct shunting from the arteries to the veins with a fast and high-pressure flow. *FSTL1* expression was significantly increased in AVM, whereas *CSPG4* showed a relatively lower increase under hypoxic stimulation because of structural abnormalities in AVM. *MARCKS* appears to be associated with angiogenesis but not AVM. Further studies are needed to improve the treatment strategies.

## Figures and Tables

**Figure 1 ijms-23-09109-f001:**
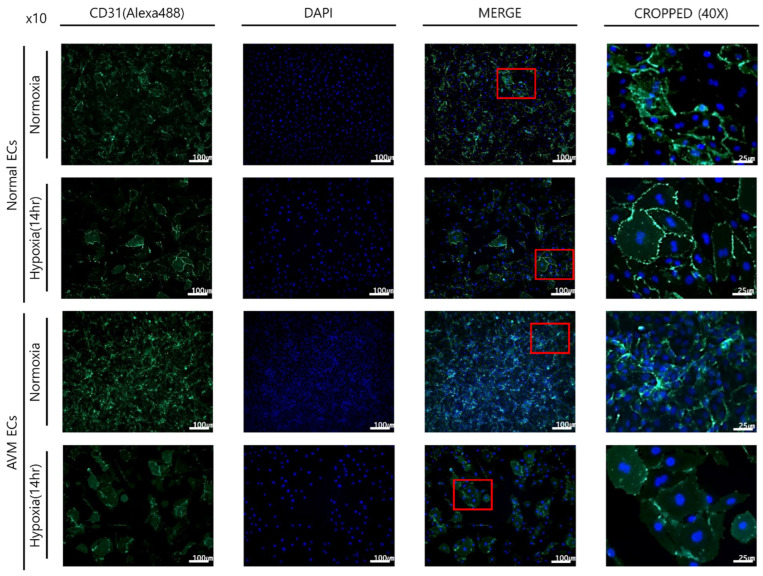
Comparison of immunofluorescence imaging after incubation under each condition for 14 h. Under normoxia, compared to that of normal ECs, AVM ECs showed higher endothelial cell proliferation. Under hypoxia, the overall proliferation of endothelial cells in both normal and AVM ECs decreased; however, proliferation increased in AVM ECs compared to that of normal ECs. AVM, arteriovenous malformation; EC, endothelial cell. Red box means Cropped (40×) image.

**Figure 2 ijms-23-09109-f002:**
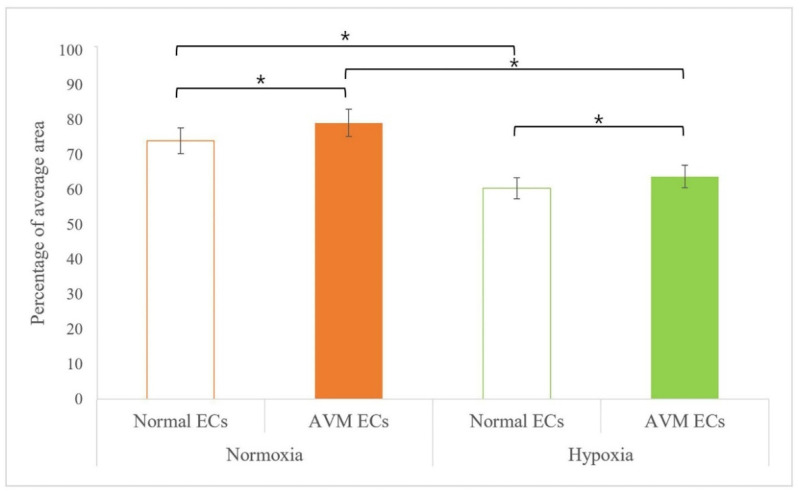
Quantitative comparison of CD31 immunoreactivity after incubation under each condition for 14 h. Pixel densities were analyzed using the ImageJ software. Compared to that of normoxia, both AVM ECs and normal ECs showed lower percentage areas of immunoreactivity in hypoxia. Compared to that of normal ECs, AVM ECs showed a higher percentage area of immunoreactivity in normoxia and hypoxia. * *p* < 0.05. AVM, arteriovenous malformation; EC, endothelial cell.

**Figure 3 ijms-23-09109-f003:**
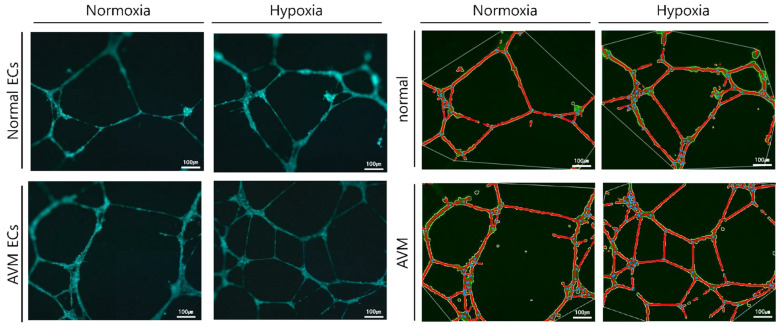
Angiogenesis assay of normal ECs and AVM ECs under normoxia and hypoxia. Tube formation assay for 8 h incubation. Vessels (red lines) and junctions (blue dots) are indicated. The total vessels lengths and total number of junctions were measured in the field. Compared with that of normal ECs, AVM ECs showed significantly increased numbers of tube junctions and vessel lengths. More and longer tubes were formed in the AVM ECs than normal ECs even in the hypoxia. AVM, arteriovenous malformation; EC, endothelial cell.

**Figure 4 ijms-23-09109-f004:**
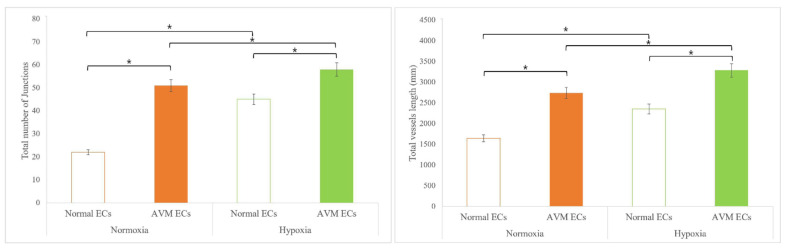
Total number of junctions and total vessel length of normal ECs and AVM ECs in normoxia and hypoxia. Compared with that of normal ECs, AVM ECs showed significantly increased total numbers of junctions and total vessel lengths. More and longer tubes were formed in AVM ECs than normal ECs even in the hypoxia. * *p* < 0.05. AVM, arteriovenous malformation; EC, endothelial cell.

**Figure 5 ijms-23-09109-f005:**
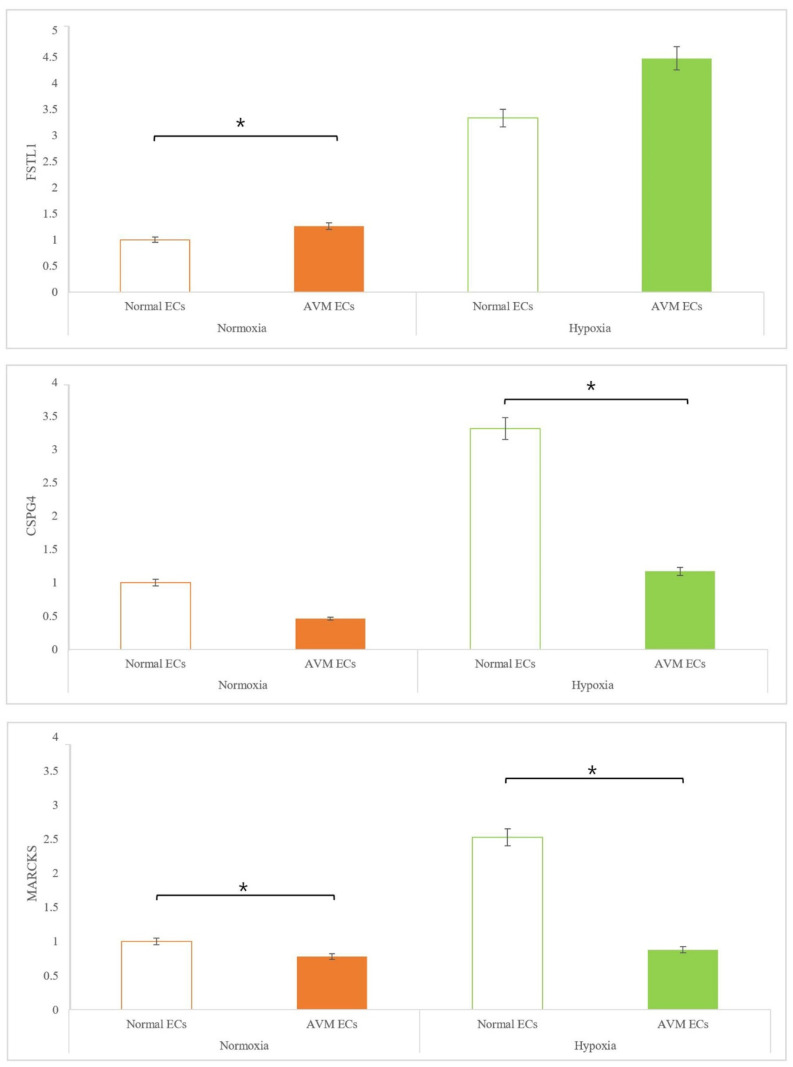
Expression levels of three target genes, *FTSL1*, *CSPG4*, *MARCKS*. Expression levels, detected using real-time PCR, were increased in hypoxia than normoxia. In hypoxia, the expression of *FSTL1* increased more than that of *CSPG4* and *MARCKS*. * *p* < 0.05. *FSTL1*, follistatin-like 1; *MARCKS*, myristoylated alanine-rich C-kinase substrate; *CSPG4*, chondroitin sulfate proteoglycan 4.

**Table 1 ijms-23-09109-t001:** The primer sequence for RT-PCR.

Gene	Primer Sequence (5′-3′)	
*GAPDH*	Forward Sequence	GGAAGGTGAAGGTCGGAGTCA
	Reverse Sequence	GTCATTGATGGCAACAATATCCACT
*FSTL1*	Forward Sequence	TCGCATCATCCAGTGGCTGGAA
	Reverse Sequence	TCACTGGAGTCCAGGCGAGAAT
*MARCKS*	Forward Sequence	CTCCTCGACTTCTTCGCCCAAG
	Reverse Sequence	TCTTGAAGGAGAAGCCGCTCAG
*CSPG4*	Forward Sequence	GTCCTGCCTGTCAATGACCAAC
	Reverse Sequence	CGATGGTGTAGACCAGATCCTC

RT-PCR, reverse transcription polymerase chain reaction; *FSTL1*, follistatin-like 1; *MARCKS*, myristoylated alanine-rich C-kinase substrate; *CSPG4*, chondroitin sulfate proteoglycan 4.

## Data Availability

The data presented in this study are available in the manuscript.

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
