# Peer review of "Hypoxia Promotes Angiogenic Effect in Extracranial Arteriovenous Malformation Endothelial Cells"

_ijms, 2022, doi:10.3390/ijms23169109_

Round 1
Reviewer 1 Report
It is very appreciable that this study was performed using clinical specimen and the limitations of sample size are recognized.
Culture expanded cells stained with fluorescent markers – why were flow cytometry methods not considered for cell identification and isolation? Line 77 mentions bioluminescence imaging?
Page 2 of 11, Line 50
It is unclear if the authors are discussing AVM as a whole or any pre-capillary shunts.
This statement is confusing since shunting could be physiological, but the ischemia and hypoxia with AVM is most likely a result of the AVM rather than the cause as mentioned at the end of that paragraph. A clarification, and rewording are requested to the first line/ statement.
Page 2 of 11, Line 67
This seems to indicate that IHC was performed on harvested tissue specimens and not on expanded cultures. Please clarify.
Page 3 of 11, Line73
A comparison of proliferation is suggested, but there are no proliferation assays presented.
No comparison is presented for total viable cells in these conditions or cells per field which many influence interpretation. No comparison of percent-positive stained cells.
Line 81: CD31 marks for EC and it is unclear how it is suggesting angiogenesis when the starting material is CD31 positive cells isolated by beads prep.
Page 4 of 11, Line 84 and Page 5 of 11 Line 107
- Fig 2 and associated description/ quantifications. This is a multiple comparison between 4 groups using a student t-test. Why was ANOVA not used? The average area of staining does not account for different cell counts and can be a confounding factor.
- In Fig 3 and associated quantification, more details are needed on the number of fields observed and used for quantification and how the fields were selected consistently across different wells and conditions. Scale bars are not shown.
- Fig 3 also suggests a more networked architecture in AVM with Hypoxia, but more rigorous quantification is essential.
- Units for network length in Fig 5 are needed. Same comments as above for multiple comparisons using t-test. Please consider rephrasing 'junctions' to prevent confusion with cell-cell junctions.
Page 8 of 11, Line 213 - 225
- The effect of a 24-hour cell extraction step by digestion on cell survival is not reported. What was cell viability after extraction and in culture and expansion conditions.
- Other factors that could influence EC responses in this model must be considered/ discussed, e.g. hypoxia in the cell isolation phase could mask EC responses, the lack of blood flow after harvest could change shear-induced EC gene expression changes.
Page 8 of 11, Line 223
Cells were isolated using a CD31 microbead kit, pre-enriching for CD31 cells and then stained with CD31 again. Did the authors consider comparison with a different marker like vWF as well?
Page 9 of 11, Line 242
CD31 expression was evaluated after a period of culture based on area of immune-reactivity.
Description of number of fields evaluated, number of cells evaluated and other details are needed.
Positive and negative control images and imaging metrics (exposure time, laser level) to minimize bias should be clarified.
Line 245, Tube formation assay
Do the authors refer to network connection nodes as 'junctions'? It might be better to choose different terminology to avoid overlap with cell-junction assessment.
Line 252 RT-PCR
It is now well established that multiple housekeeping gene normalization technique is superior to single housekeeping gene normalization for simple PCR comparisons. Example publication: https://doi.org/10.1007/s11684-008-0045-7
Were other housekeeping genes considered, or was it confirmed that GAPDH as a housekeeping gene does not show significant variations under test conditions?
In Fig 5, comparisons are suggested across genes as well. The analysis techniques must be clarified and explained in detail?
An attenuation of gene expression response in 2 of the 3 targets compared to the "normal" tissue could potentially be explored by using a normalization approach (i.e. response compared to response of Normal tissue) to clarify interpretations and to reduce number of comparisons.
Line 263 Statistical test
In several of the comparisons, there are more than 2 groups evaluated and no correction of p value to account for these groups is presented neither is a reasoning for not using ANOVA.
Fig 4 on Page 5, Line 106, shows inferences of significance drawn across the test groups (Hypoxia/Normoxia) and pathology (AVM/Normal)
Were ANOVA and similar tests for more than 2 groups considered?
Was as simplification of comparisons to a normalized ratio considered?
Author Response
We corrected our article with professional English correction once again. Thank you for reviewing our article.

Reviewer 2 Report
Comments:
1. It would be clear if the authors explain briefly, how the genes were selected for the mRNA expression analysis between the groups in the ECs.
2. Throughout the manuscript, scale bars were missing in all the microscopic images. Please update the images with accurate scale. This will be very important in the accuracy of comparative analysis of these images.
3. References are missing in various places as mentioned. Line 45-49, 50-54.
4. Rewrite the sentence in line 228-229.
5. In the method section line 220, where the authors describe the culture and isolation of ECs, please carefully check the name of the growth medium. Is this EMB-2 or EBM-2 medium?
6. In the method section line no 240-241, the authors described that they acquired the images with the help of Zeiss fluorescence microscope the images were analyzed using image J software. However, in the result section, they have also mentioned that the bioluminescence image was also acquired. Please confirm this and there are no bioluminescence images provided in the figure.
7. In line 67 “Immunocytochemistry performed on ECs isolated from human normal tissues or AVM tissues. This opening statement of the result can be improved. Please re-write this sentence.
8. The authors concludes that the proliferation was more in AVM as compared to the normal ECs. Please confirm how was the proliferation checked in this ECs? The authors shall performed appropriate experiments to support these statement. For instances, please consider MTS assay or other proliferation assay (PCNA, Ki67 etc staining).
9. The authors describe that “although there was no marked difference in the expression level of MARCKS under normoxia and hypoxia, MARCKS expression was significantly decreased in AVMs and normal tissues”. This statement is confusing as the figure suggested different expression level in MARCKS in normal and hypoxia in both normal and AVMs. Please check.
10. The authors wrote in their introduction as “Anti-angiogenic drugs such as vascual endothelial growth factor (VEGF) blockers are also used as a treatment to prevent the vicious cycle of regeneration ………………. Please mention what are the other alternatives used for treatment. Also, please correct the spelling mistake here. VEGF is spelled incorrectly.
11. Also, the authors wrote that “Moreover, various studies reported that angiogenesis occurs under hypoxia based on evaluation of CD31 expression [3,4]. Please clarify, it is true for both normal and pathophysiological situations.
Author Response

(The authors gave the same response as above.)

Round 2
Reviewer 2 Report
The authors have addressed all the comments and concerns. At this stage there is no further concerns and the manuscript is found to be suitable for the editor's consideration